# Hyperthermic Intravesical Chemotherapy (HIVEC) Using Epirubicin in an Optimized Setting in Patients with NMIBC Recurrence after Failed BCG Therapy

**DOI:** 10.3390/cancers16071398

**Published:** 2024-04-02

**Authors:** Julien Blanc, Jonathan Ruggiero, Ilaria Lucca, Nicolas Arnold, Bernhard Kiss, Beat Roth

**Affiliations:** 1Department of Urology, University Hospital of Lausanne, University of Lausanne, 1011 Lausanne, Switzerland; julien.blanc@chuv.ch (J.B.); jonathan.ruggiero@chuv.ch (J.R.); ilaria.lucca@chuv.ch (I.L.); 2Department of Urology, University Hospital of Bern, University of Bern, 3010 Bern, Switzerland; nicolas.arnold@insel.ch (N.A.); bernhard.kiss@insel.ch (B.K.)

**Keywords:** bladder cancer, lymph node dissection, cystectomy, gastrointestinal recovery, complications, postoperative pain

## Abstract

**Simple Summary:**

Intravesical instillation using bacillus Calmette-Guérin (BCG) is the standard therapy for patients with high-risk, non-muscle-invasive bladder cancer. If BCG therapy fails, however, radical cystectomy with urinary diversion is recommended. Still, some patients are unfit or just refuse major surgery. We showed in this study that for these patients at highest risk for recurrence and progression, device-assisted hyperthermic intravesical chemotherapy using conductive heating and epirubicin in an optimized setting might be a bladder-sparing alternative with excellent recurrence-free survival rates within the bladder and without severe side effects. Still, a major concern is the extravesical manifestation of urothelial cancer in these patients with poor prognosis due to the associated risk of metastasis. Urologists should be aware that the window of opportunity for cure by undergoing radical cystectomy is tight and should not be missed.

**Abstract:**

To evaluate hyperthermic intravesical chemotherapy (HIVEC) using conductive heating and epirubicin in an optimized setting as an alternative to radical cystectomy in patients with recurrent non-muscle invasive bladder cancer (NMIBC) who have failed bacillus Calmette-Guérin (BCG) therapy. We retrospectively analyzed our prospectively recorded database of patients who underwent HIVEC between 11/2017 and 11/2022 at two Swiss University Centers. Cox regression analysis was used for univariate/multivariate analysis, and the Kaplan–Meier method for survival analysis. Of the 39 patients with NMIBC recurrence after failed BCG therapy, 25 (64%) did not recur within the bladder after a median follow-up of 28 months. The 12- and 24-month intravesical RFS were 94.8% and 80%, respectively. Extravesical recurrence developed in 14/39 (36%) of patients. Only 7/39 (18%) patients had to undergo radical cystectomy. Seven patients (18%) progressed to metastatic disease, with five of these (71%) having previously developed extravesical disease. No adverse events > grade 2 occurred during HIVEC. Device-assisted HIVEC using epirubicin in an optimized setting achieved excellent RFS rates in this recurrent NMIBC population at highest risk for recurrence after previously failed intravesical BCG therapy. Extravesical disease during or after HIVEC, however, was frequent and associated with metastatic disease and consecutively poor outcomes.

## 1. Introduction 

Bladder cancer (BC) is one of the ten most diagnosed cancers worldwide [1]. In 70% of cases, it is diagnosed as non-muscle-invasive bladder cancer (NMIBC), suggesting favorable oncological outcome. The recurrence rates of high-grade NMIBC, however, are as high as 15–61% and 31–78% at one and five years, respectively, even when the tumor is fully resected by TURBT. Even more important, the progression rate is up to 17% and 45% at one and five years, respectively [2]. Due to its heterogeneity, the European Organization for Research and Treatment of Cancer (EORTC) created a scoring model that allows for the classification of NMIBC patients into different risk groups according to their prognosis score of recurrence and progression. For patients with intermediate or high-risk NMIBC, intravesical immunotherapy with Bacillus Calmette-Guérin (BCG) is considered the most effective adjuvant treatment after complete resection [3,4,5,6]. Still, up to 30–40% of patients [7] experience recurrent disease (BCG-relapsing), while others are BCG-unresponsive, showing a persistence of disease (BCG-refractory) or a development of high-grade tumor within 6 mts or CIS within 12 mts of adequate BCG treatment [8]. These patients are unlikely to respond to further BCG treatment, and radical cystectomy (RC) should be offered [8]. 

Some patients, however, are unfit to undergo cystectomy, while others are just unwilling to undergo major surgery with long-lasting consequences (e.g., erectile dysfunction, incontinent urinary diversion with an ileostoma) for “superficial” cancer. Thus, the use of alternative intravesical therapies is expanding worldwide [9,10,11,12,13,14]; among these, the combination of a cytotoxic agent (mostly Mitomycin C [MMC]) and hyperthermia delivered intravesically (device-assisted hyperthermic intravesical chemotherapy; HIVEC) has been explored since the late 1990s. Hyperthermia between 40 and 44 °C enhances the cytotoxic effects of chemotherapeutic agents [15]. Although HIVEC using MMC and a microwave-induced system that heats the bladder wall has shown promising oncological results [16], the treatment may provoke frequent side effects and even severe adverse effects, leading to the discontinuation of treatment in up to 38% of cases [17]. 

To overcome these drawbacks, we prospectively evaluated the combination of an alternative delivery system that uses conductive heating (Unithermia^®^,Elmedical Ltd. Hod-Hasharon, Israel) [18] with an alternative chemotherapeutic (epirubicin; an anthracycline neoplastic agent that inhibits DNA replication, transcription, and repair) in an optimized setting in patients with high-grade NMIBC recurrence or persistence after/during BCG treatment (BCG failure). 

## 2. Patients and Methods

This is a retrospective analysis of a prospective, observational, investigator-initiated, and institutional board-reviewed study (Local Ethics Committee, Switzerland; protocol number 121/08) conducted in two Swiss University Hospitals (Bern and Lausanne). From 11/2017 to 11/2022, 39 patients with high-grade NMIBC recurrence or persistence following/during BCG therapy were treated with HIVEC according to an institutional protocol; 95% of these patients (*n* = 37) had refused cystectomy, and 5% (*n* = 2) were unfit for major surgery. The study has been performed in accordance with the ethical standards laid down in the 1964 Declaration of Helsinki and its later amendments. All patients gave their informed consent prior to their inclusion in the study.

### 2.1. Patients

All patients were BCG failures: They had pathologically proven high-grade NMIBC occurring following 6 cycles of BCG induction or during/following BCG maintenance therapy (BCG-relapsing: *n* = 22; BCG-unresponsive: *n* = 17 (of which 13 were BCG-refractory). Median time from last BCG treatment was 12 mths (IQR: 6–28; Table 1). Diagnosis was made by trans-urethral bladder tumor resection (TURBT) or bladder mapping biopsies in cases of a positive cytology and absence of macroscopic tumor at white light and/or blue light (Hexvix^®^) cystoscopy. Patients with pT1 or high-grade pTa tumors had a second transurethral resection 2–4 weeks after the first TURBT to exclude muscle-invasive or residual disease (except Cis). Extravesical disease before HIVEC was excluded by biopsies of the prostatic urethra, selective cytologies, and contrast-enhanced CT of the upper urinary tract (UUT). All patients were at high or very high risk for progression to muscle-invasive disease according to the EAU risk groups [4]. Exclusion criteria were muscle-invasive disease, histology other than urothelial, known allergy/intolerance to epirubicin, or if the patients did not consent to the HIVEC protocol.

### 2.2. Treatment 

The device-assisted HIVEC was performed using the UniThermia^®^ heating system with epirubicin (50 mg diluted in 50 mL of 0.9% saline) heated to 43 °C for 50 min. Patients were treated during the 12 h preceding the HIVEC with 4 g of sodium bicarbonate (NaBic) to alkalinize the urine (target urinary pH 6 or greater), as urine pH has been shown to have a marked influence on drug activity, with epirubicin being the more cytotoxic the higher the urine pH [19]. NaBic medication was adjusted after each treatment individually according to the measured urine pH values. Furthermore, patients were not allowed to drink more than 2 dL of fluids (no coffee at all) within the 4 h preceding the HIVEC to keep drug dilution as low as possible [19,20].

#### Schedule

Treatment was administered as an induction 6 times weekly and as a maintenance 6 times monthly thereafter (total duration of therapy: 9 mts). Patients underwent follow-up cystoscopy and cytology every 3 months for 3 years (starting 3–6 weeks after the induction cycle), and every 6 months thereafter for a total of at least 5 years if they remained tumor-free. In case of a suspected bladder cancer recurrence on cystoscopy, a TURBT was performed for histological confirmation. If cytology revealed high-grade recurrence without cystoscopic evidence of recurrence, transurethral random biopsies of the bladder and the prostatic urethra, as well as selective cytologies of the upper urinary tract (UUT; with diagnostic ureteroscopy if deemed necessary), were performed. Blue-light cystoscopy was added according to the urologist’s preference. Additionally, computerized tomography (CT) of the abdomen was performed after 6, 12, 24, and 60 months, and whenever cystoscopy and/or cytology revealed disease recurrence. 

### 2.3. Outcome Measurements

The primary outcome was recurrence-free survival (RFS). Tumor recurrence was defined as any histologically proven urothelial carcinoma of the bladder of any stage and grade (including pTa low-grade disease). Secondary outcomes were progression-free (PFS; defined as a muscle-invasive disease and/or development of metastasis at imaging), cancer-specific (CCS) and overall survival (OS), the rate of extravesical disease recurrence (UUT or prostatic urethra), treatment discontinuation (<5 induction and/or <3 maintenance cycles administered), and the need to perform an RC. Adverse events were graded using the Common Terminology Criteria for Adverse Events (CTCAE) v4.03. For further analysis, the adverse events were grouped into medically non-significant (severe/mild: grade 1–2), medically significant (severe/life threatening: grade 3–4), and lethal (grade 5). 

### 2.4. Statistical Analysis

Qualitative data are presented as counts and percentages. Cox regression analysis (univariate) was performed to evaluate different variables (EAU risk category [high or highest], EORTC recurrence and/or progression score [quantitative], presence of CIS [yes/no], presence of extravesical disease [yes/no], total number of HIVEC instillations administered (induction and maintenance), the number of recurrences before HIVEC, tumor stage, patient age, and gender), and the incidence of NMIBC recurrence and progression. For the multivariate Cox regression model, all variables with a *p*-value ≤ 0.05 in univariate regression analysis were selected. Results were presented as hazard ratios (HR) and 95% confidence intervals (CI). Survival analysis was performed using the Kaplan–Meier method. A two-tailed level of significance was set at 0.05 for all statistical tests. SPSS statistical software package 20.0 (SPSS, Inc., Chicago, IL, USA) was used for all statistical analyses.

## 3. Results

Of the 39 patients (33 male (85%), median age: 70 yrs (range: 44–93; Table 1) included in the trial, 8 (21%) discontinued treatment early: 4 due to side effects, 2 due to tumor progression, and 2 due to persistence of Cis after 3 and 6 months despite adequate therapy. A median of 8 (IQR: 6–12) HIVEC cycles per patient were administered. Besides the previous BCG instillations, eight patients had additionally received at least 1 previous cycle (3 or more instillations) of intravesical chemotherapy (mitomycin C or epirubicin; not heated) before entering this prospective trial.

After a median follow-up of 28 months (range: 4–59), 25/39 (64%) patients showed no recurrence within the bladder. The pathology of the 14/39 (36%) patients that had an intravesical BC recurrence revealed persistent or recurrent Cis in 5/14 patients, pTa low grade in 1/14, pTa high grade in 2/14, pT1 in 2/14, and muscle-invasive disease (≥pT2) in 4/14. Median time to recurrence was 13 mts (IQR: 6–22). The 12- and 24-month intravesical RFS was 94.8% and 80%, respectively *(*Figure 1A).

Fourteen of 39 (36%) patients developed extravesical disease: 9 in the UUT, 1 in the prostatic urethra, and 4 in both the UUT and the prostatic urethra after a median follow-up of 9 mts (IQR: 6–18). The pathology of the extravesical urothelial cancer recurrence showed Cis in 6, pTa in 7, and ≥pT1 in 4 patients. Of the 14 patients who developed a BC recurrence, 5/14 (36%) patients finally had additional extravesical disease, leaving 9/36 (25%) of patients with a urothelial cancer recurrence that was only located within the bladder.

Seven of 39 (18%) patients had to undergo RC due to recurrent/persistent intravesical HG NMIBC (*n* = 4), due to intravesical tumor progression to muscle-invasive disease (*n* = 2), or due to contracted bladder without any signs of intravesical tumor recurrence (*n* = 1). Seven patients (18%) finally progressed to metastatic disease; four of these seven patients had high-grade extravesical disease only (UUT: *n* = 3; prostatic urethra and UUT: *n* = 1), 1 patient had muscle-invasive bladder cancer (MIBC) recurrence and concomitant extravesical recurrence of the prostatic urethra (>1), and only 2 patients had HG urothelial cancer recurrences restricted to the bladder. PFS (no muscle-invasive disease, no metastases) is shown in Figure 1B. Ten of 39 (26%) patients died; 2 bladder cancer related, 5 extravesical urothelial cancer related, and 3 due to other causes (Appendix A).

Univariate analysis revealed that Cis at HIVEC (HR: 8.45; CI: 1.03–69.30) and higher EORTC recurrence category (HR: 4.83; CI: 1.15–20.30) were independent risk factors for intravesical recurrences. Since Cis is part of the EORTC recurrence score, multivariate analysis did—as expected—reveal that the two variables were not independently associated with cancer recurrence (Table 2).

As for cancer progression to muscle-invasive or metastatic disease, no variable was statistically significantly associated with a higher risk of progression, mainly due to the low number of events; still, a trend was seen for the EORTC progression categories (*p* = 0.11, Table 3). We did not find any factors that were significantly associated with the occurrence of extravesical disease of urothelial cancer.

Side effects/adverse events were usually mild/moderate and locally limited to the bladder; bladder spasms/pain grade 1–2 occurred in 15% (7/39) of patients at least once during the 9-month treatment period, hematuria grade 1–2 in 15% (6/39), and urgency/frequency grade 1–2 in 23% (9/39). Two of these nine patients with urgency/frequency syndrome were diagnosed with eosinophilic cystitis, diagnosed with cystoscopy and biopsy under local anesthesia (grade 2). Urinary tract infections (UTIs) were rather frequent: 27/39 patients (69%) experienced at least one symptomatic UTI during their HIVEC requiring oral antibiotics (grade 2; no hospitalization or i.v. antibiotic administered). No systemic adverse events typically associated with epirubicin given intravenously (such as nausea, diarrhea, bone marrow depression, hair loss, fatigue, or allergic reaction) were observed. No patient experienced any severe adverse grade 3–4 or lethal (grade 5) event.

## 4. Discussion

Patients with BCG-relapsing and especially BCG-unresponsive tumors are at risk for tumor progression to systemic disease with inherently reduced disease-specific survival [21]. Early radical cystectomy is therefore recommended in these patients [8] to achieve optimal oncologic disease control. Some patients, however, are not willing or unfit to undergo major surgery. Thus, alternative treatments are investigated. To be able to rank all these emerging therapy alternatives based on efficacy and clinical importance, the IBCG [22] considered “complete response rates (carcinoma in situ; CIS) and recurrence-free rates (papillary tumors) of 50% at 6 months, 30% at 12 months, and 25% at 18 months as clinically meaningful”, while a public workshop of the FDA and the American Urological Association (AUA) proposed 40–50% at 6 months and 30% at 18–24 months (regardless of CIS or papillary tumor) [23]. A recent study using a discrete event simulation framework in combination with a supercomputer proposed to “increase these thresholds to at least 45–55% at 6 months and 35% at 18–24 months (complete response rates/recurrence-free survival) to promote the development of clinically truly meaningful novel therapies” [24]. In this context, our 80% of recurrence-free survival at 24 months for a second line “conservative” treatment of high-grade NMIBC after previous failed BCG therapy appears to be very favorable and almost three times as high as the minimal value requested by the IBC.

The results of our device-assisted HIVEC using epirubicin and conductive heating are even more impressive, as 36% of patients with BC recurrence had in fact concomitant extravesical disease, which may have caused BC recurrence after termination of HIVEC. The reasons for these excellent results are likely to be manifold: First, the circulation of the drug by pump allows optimal distribution within the bladder and thus optimal mucosal contact. Second, not only an induction but also a maintenance therapy was carried out. Third, and probably most important, was the optimal setting with fluid restriction and pH adjustment to ≥6.5, which significantly increases the effect of epirubicin [19,20].

While RC offers maximal local disease control, it represents a procedure that carries a risk of 90-day major complications of 11% to 35% and a risk of mortality of 3% to 5% [25]. This is clearly in contrast to the device-assisted HIVEC using epirubicin presented in this trial, in which no severe adverse event ≥ grade 3 was reported. Local side effects (grade 1–2), including UTIs, were relatively common but usually easy to treat. This low risk of severe side effects is important when an alternative, bladder-sparing therapy is considered; the ratio of benefit vs. serious adverse events/toxicity grade 3–4 should be as favorable as possible (at best > 3) to justify an alternative treatment approach and to have maximal patient compliance. In this context, the 2-year CSS of 97.3% in our trial is equal to the 2-year CSS of 93% after RC in patients with <pT2 BC recurrence after previous BCG therapy [26].

Importantly, 36% of patients developed extravesical disease in the UUT and/or the prostatic urethra relatively shortly (median: 9 mts) after the beginning of the HIVEC. This is even more astonishing, as all patients had a thorough evaluation of the UUT (CT, selective cytologies) as well as (in men) the prostatic urethra (including biopsies) prior to the HIVEC. Yet, it shows that urothelial cancer is a panurothelial disease and confirms once again the fact that BCG failure is often the failure of the urologist (collectively with the radiologist) to detect extravesical urothelial disease [27].

Seven patients (18%) progressed to metastatic disease; however, five of these seven patients presented with high-grade extravesical urothelial cancer during or after HIVEC, and four of these five patients did not have concomitant BC recurrence. Thus, metastases were rather due to the extravesical than the bladder manifestation of the urothelial cancer; the origin of the urothelial metastases, however, could not be assessed conclusively as no molecular analysis was performed. Still, this confirms that the occurrence of concomitant urothelial cancer within the UUT [28] and/or prostatic urethra [29] is associated with worse prognostic outcome than purely intravesical disease; probably—at least partially—due to a delay in diagnosis. It further points out that recurrent high-grade NMIBC that does not respond to BCG therapy is extremely aggressive and has a relevant potential for metastasis. The window of opportunity for cure by a more aggressive therapy such as RC should therefore not be missed [19,20,21].

As shown by van den Bosch et al. [30], the progression to MIBC and bladder-cancer-related death in a high-risk NMIBC population is usually early, occurring mainly within 48 months. This is in line with the RC series after BCG failure, showing an initial drop of CSS within the first 24 mts with a clear flattening of the curve afterwards. In our study, tumor progression, if present, also occurred relatively early (median: 15 mts, IQR: 6–22). Still, a relatively short median follow-up of 28 months does not allow to draw further conclusions, although in the context of the aforementioned literature, it can be assumed that the rate of tumor progression should not increase too much during further follow-up. In this context, we argue that patients must be closely followed-up during the first 2 years after initiating HIVEC to detect tumor progression in time. Unfortunately, it is often difficult—even with proven recurrence and/or progression—to convince patients who demand non-invasive therapies to undergo major surgery (RC).

To further evaluate risk factors for tumor recurrence and progression after HIVEC for NMIBC recurrence after previous BCG therapy, we performed a Cox-regression analysis of multiple variables, showing that the EORTC scores could best predict recurrence and progression, respectively. Interestingly, extravesical disease was not associated with progression or recurrence of NMIBC to MIBC within the bladder itself in our trial (Table 2 and Table 3), although it has a poor prognosis in terms of metastasis and therefore CSS. Thus, we strongly recommend using the EORTC scoring calculator to better select patients who are potential candidates for a bladder sparing treatment approach, and especially to convince those patients who are not.

The study is not without any limitations; besides the retrospective single-arm design, its limited number of patients makes a thorough analysis of risk factors for recurrence and progression difficult. Still, EORTC risk categories turned out to be rather predictive, which renders the data trustworthy. Furthermore, the population is quite heterogeneous, particularly in terms of delay after BCG therapy and pathological characteristics of the recurrence before HIVEC. Still, the well-defined protocol with induction and maintenance cycles, the prospective collection of data including patient questionnaires before and after each therapy, the multicentric nature, the good compliance without any high-grade adverse events, but especially the excellent intravesical RFS at 2 years of 80% for a population of NMIBC patients at high/highest risk for recurrence and progression make this study clinically meaningful.

## 5. Conclusions

In a population with high/highest risk NMIBC recurrence after previous intravesical BCG therapy, device-assisted HIVEC using epirubicin as induction and maintenance in an optimized setting achieved excellent intravesical RFS rates of 94.8% and 80% at 1 and 2 years, respectively, and enabled bladder preservation in 82% of patients. The ratio of benefit vs. serious adverse events and toxicity was very favorable. The risk of progression to muscle-invasive and/or metastatic disease, however, was not negligible; the latter was mainly associated with extravesical manifestation of the disease during or after HIVEC. Urologists should be aware that the window of opportunity for cure by undergoing radical cystectomy is tight and should not be missed; still, this opportunity may already be gone in some patients when it comes to even discussing bladder preservation alternatives.

## Figures and Tables

**Figure 1 cancers-16-01398-f001:**
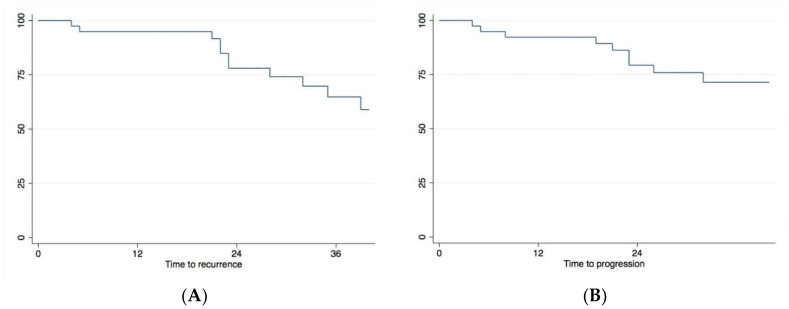
(**A**) Recurrence-free (RFS) and (**B**) progression-free survival (PFS) following device-assisted HIVEC for NMIBC recurrence after failed BCG therapy using conductive heating of epirubicin in an optimized setting.

**Table 1 cancers-16-01398-t001:** Patients’ characteristics.

Variable	
Number of patients included	39
Follow-up—median (IQR)	28 (20–39)
Age—median; years (IQR)	70 (64–77)
Months after BCG treatment—median (IQR)	12 (6–28)
Median number of recurrences before HIVEC—n (range)	2 (1–7)
Gender—n (%)	
Female	6 (15)
Male	33 (85)
Tumor stage at HIVEC (stage of recurrence after previous BCG therapy—n (%)	
pTa high grade	13 (33)
pT1	13 (33)
pTis only	13 (33)
Highest tumor stage at any time before HIVEC—n (%)	
pTa high grade	13 (33)
pT1	24 (62)
pTis only	2 (5)
BCG failure—n (%)	
BCG refractory	13 (33)
BCG relapsing	22 (57)
BCG unresponsive	4 (10)
EORTC recurrence score category—n (%)	
1 (score 1–4)	2 (5)
2 (score 5–9)	28 (72)
3 (score 10–17)	9 (23)
EORTC progression score category—n (%)	
1 (score 2–6)	1 (2)
2 (score 7–13)	17 (44)
3 (score 14–23)	21 (54)
EAU risk category [8]—n (%)	
High	13 (33)
Very high	26 (67)

**Table 2 cancers-16-01398-t002:** Univariate and multivariate analysis predicting bladder recurrence after HIVEC.

Variable	Univariate			Multivariate		
	HR	95% CI	*p*	HR	95% CI	*p*
Age	0.98	0.93–1.04	0.54			
Gender	2.77	0.31–24.88	0.31			
BCG unresponsive (yes/no)	3.51	0.76–16.31	0.11			
Number of HIVEC instillations	0.61	0.26–1.46	0.27			
Tumor stage	1.90	0.89–4.06	0.10			
CIS at any time before HIVEC	1.10	0.24–5.09	0.91			
CIS at HIVEC	8.45	1.03–69.30	0.047	2.52	0.11–57.36	0.56
Number of recurrences before HIVEC	0.74	0.46–1.19	0.21			
Months after BCG instillation	0.93	0.87–1.01	0.07			
EORTC recurrence score	4.83	1.15–20.30	0.031	2.85	0.34–23.53	0.33
EAU risk category (high/highest)	0.57	0.17–1.95	0.38			
Extravesical disease	1.38	0.46–4.14	0.56			

**Table 3 cancers-16-01398-t003:** Univariate analysis predicting disease progression after HIVEC.

Variable	Univariate		
	HR	95% CI	*p*
Age	1.05	0.98–1.12	0.15
Gender	1.69	0.19–15.25	0.64
BCG unresponsive (yes/no)	3.98	0.55–28.72	0.17
Number of HIVEC instillations	0.71	0.40–1.28	0.26
Tumor stage	0.72	0.33–1.61	0.43
CIS at any time before HIVEC	1.03	0.48–2.22	0.93
CIS at HIVEC	1.64	0.40–6.75	0.50
Number of recurrences before HIVEC	0.75	0.49–1.15	0.19
Months after BCG instillation	0.97	0.93–1.01	0.18
EORTC progression score	1.27	0.94–1.72	0.11
EAU risk category (high/highest)	2.57	0.54–12.16	0.23
Extravesical disease	1.70	0.49–5.93	0.40

## Data Availability

The data presented in this study are available on request from the corresponding author.

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
