# Peer review of "Hyperthermic Intravesical Chemotherapy (HIVEC) Using Epirubicin in an Optimized Setting in Patients with NMIBC Recurrence after Failed BCG Therapy"

_cancers, 2024, doi:10.3390/cancers16071398_

Round 1

Reviewer 1 Report

Comments and Suggestions for Authors

This is a well written paper on HIVEC with Epirubicin on a dicentric cohort of 39 patients with NMIBC declining or not eligible for cystectomy. The paper is informative and describes the positive value of HIVEC as an alternative for difficult clinical situations broadening the therapeutic options. This option is sufficiently clinically assessed. It may be added that the value is increases if non-invasive high grade lesions are considered for HIVEC ( CIS and pTaHG). The paper is well publishable, however would rather interest the clinical urologist, thus probably be preferable for a urology journal?

Line 126/127: the Definition of PFS cases is described as progression cases, not as progression free. 

Author Response

This is a well written paper on HIVEC with Epirubicin on a dicentric cohort of 39 patients with NMIBC declining or not eligible for cystectomy. The paper is informative and describes the positive value of HIVEC as an alternative for difficult clinical situations broadening the therapeutic options. This option is sufficiently clinically assessed. It may be added that the value is increases if non-invasive high grade lesions are considered for HIVEC ( CIS and pTaHG). The paper is well publishable, however would rather interest the clinical urologist, thus probably be preferable for a urology journal?

We thank the reviewer for the comment. However, as it is the special issue “Advances in Management of Urothelial Cancer» we are conmvinced that the paper perfectly fits into the concept of the journal (and it special issue).

Line 126/127: the Definition of PFS cases is described as progression cases, not as progression free.

While progression was a major issue, we think that the common definition of progression-free survival is rather appropriate as it allows to show the standard Kaplan-Meier curves. We thus kept the wording.

Reviewer 2 Report

Comments and Suggestions for Authors

It's an interesting work that addresses a clinical need more relevant today than ever. The topic of salvage therapy in BCG failure/unresponsive cases is particularly timely, considering that the future direction will no longer be cystectomy but medical therapies.

No improvement in the methodology is required. Optimal control would be with a "radical cystectomy" arm, but we well know that it is not possible for clinical and ethical reasons within the context of evaluating the efficacy of bladder sparing strategies. Hence the authorization of single-arm trials. Moreover, 95% of these patients refused cystectomy and 5% were unfit. The conclusions are consistent with the results presented, summarizing the key points and addressing an important clinical need. The references are appropriate. The quality of the data is good, and the results are in line with the current literature. Tables and figures are well constructed and explanatory. The only comment is on table 2, even if no significance is found at the univariate level for certain variables, these should still be included (and therefore the data reported) in multivariate analysis.

Author Response

It's an interesting work that addresses a clinical need more relevant today than ever. The topic of salvage therapy in BCG failure/unresponsive cases is particularly timely, considering that the future direction will no longer be cystectomy but medical therapies.

No improvement in the methodology is required. Optimal control would be with a "radical cystectomy" arm, but we well know that it is not possible for clinical and ethical reasons within the context of evaluating the efficacy of bladder sparing strategies. Hence the authorization of single-arm trials.

We completely agree with the reviewer that a randomized trial including cystectomy would have been the best trial to clearly show the differences in survival rates between cystectomy and bladder sparing therapies. However, as pointed out by the reviewer, this will be a mission impossible from the ethical point of view.

Moreover, 95% of these patients refused cystectomy and 5% were unfit. The conclusions are consistent with the results presented, summarizing the key points and addressing an important clinical need. The references are appropriate. The quality of the data is good, and the results are in line with the current literature. Tables and figures are well constructed and explanatory. The only comment is on table 2, even if no significance is found at the univariate level for certain variables, these should still be included (and therefore the data reported) in multivariate analysis.

We thank the reviewer for their comment. As for table 2, no analysis was performed with variables that showed no significant difference in univariate analysis as stated in the method section: “For the multivariate Cox regression model, all variables with a p-value ≤ 0.05 in univariate regression analysis were selected”. This definition was chosen as the number of events was too small to include all variables in a multivariate analysis.
